# Understanding the Thermal Runaway of Ni-Rich Lithium-Ion Batteries

**Thi Thu Dieu Nguyen** [1,2,3,*] 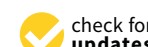**, Sara Abada** [1]**, Amandine Lecocq** [2]**, Julien Bernard** [1]**, Martin Petit** [1]**, Guy Marlair** [2]**, Sylvie Grugeon** [3,4] **and Stéphane Laruelle** [3,4]

[1]   IFP Energies Nouvelles, Rond-Point de L'échangeur de Solaize—BP3, 69360 Solaize, France; sara.abada@ifpen.fr (S.A.); julien.bernard@ifpen.fr (J.B.); martin.petit@ifpen.fr (M.P.)

[2]   INERIS, Parc Technologique Alata—BP 2, 60550 Verneuil-En-Halatte, France; amandine.lecocq@ineris.fr (A.L.); guy.marlair@ineris.fr (G.M.)

[3]   Laboratoire de Réactivité et Chimie des Solides, CNRS UMR 7314, Université de Picardie Jules Verne, 33 rue Saint Leu, 80039 Amiens, France; sylvie.grugeon@u-picardie.fr (S.G.); stephane.laruelle@u-picardie.fr (S.L.)

[4]   Réseau Sur le Stockage Electrochimique de l'Energie-RS2E-CNRS FR3459, 33 rue Saint Leu, 80039 Amiens, France

*   Correspondence: nthithudieu@gmail.com; Tel.: +33-768-343-744

**Abstract:** The main safety issue pertaining to operating lithium-ion batteries (LIBs) relates to their sensitivity to thermal runaway. This complex multiphysics phenomenon was observed in two commercial 18650 Ni-rich LIBs, namely a Panasonic NCR GA and a LG HG2, which were based on $Li(Ni_{0.8}Co_{0.15}Al_{0.05})O_2$ (NCA) and $Li(Ni_{0.8}Mn_{0.1}Co_{0.1})O_2$ (NMC811), respectively, for positive electrodes, in combination with graphite-SiO$_x$ composite negative electrodes. At pristine state, the batteries were charged to different levels of state of charge (SOC) (100% and 50%) and were investigated through thermal abuse tests in quasi-adiabatic conditions of accelerating rate calorimetry (ARC). The results confirmed the proposed complete thermal runaway of exothermic chain reactions. The different factors impacting the thermal runaway kinetics were also studied by considering the intertwined impacts of SOC and the related properties of these highly reactive Ni-rich technologies. All tested cells started their accelerated thermal runaway stage at the same self-heating temperature rate of ~48 °C/min. Regardless of technology, cells at reduced SOC are less reactive. Regardless of SOC levels, the Panasonic NCR GA battery technology had a wider safe region than that of the LG HG2 battery. This technology also delayed the hard internal short circuit and shifted the final venting to a higher temperature. However, above this critical temperature, it exhibited the most severe irreversible self-heating stage, with the highest self-heating temperature rate over the longest duration.

**Keywords:** lithium-ion battery; safety; thermal runaway; Ni-rich; energy storage

---

## 1. Introduction

Lithium-ion battery (LIB) is one of the most important energy storage technologies available today, thanks to their high specific energy densities and stable cycling performance [1,2]. The challenging requirements for LIB technology are (i) the targeting of lower cost systems, (ii) achieving higher performance with a longer lifetime (>10 years for automotive applications), (iii) allowing fast-charging (<20 min for 80% state of charge (SOC)), and (iv) providing low temperature cycling. At the same time, these expected improvements should not compromise safety performance, which must remain excellent in all situations (i.e., over the whole lifetime, including in all weather and abuse conditions) [3].

However, LIBs contain flammable electrolytes, and thus not only do they store electrical energy in the form of chemical potential energy, they also store chemical energy (especially compared to

cells with water-based electrolytes) in the form of combustible materials [4]. Therefore, when LIBs are operated improperly, either outside of the specifications of its manufacturer or due to cell defects, electrical and chemical energies inside the cells can be unintentionally released and lead to gassing, fires, or even explosions. During these incidents, the most energetic catastrophic failure of a LIB system is a cascading thermal runaway event. This is characterized by a deficit of energy evacuation versus energy accumulation in the cells, leading to uncontrollable overheating of the battery system. In general, this energetic failure occurs when an exothermic reaction gets out of control. As the temperature of the battery rises to a certain threshold, the exothermic chemical reaction rate inside the battery increases and further heats up the cell. The continuously rising temperatures may trigger cascading chain reactions [3,5] and result in uncontrolled flammable and toxic gassing, fires, and explosions, which are especially critical for large battery packs.

Since the commercialization of LIBs by Sony Inc. in 1991 until today, recurrent incidents involving LIBs undergoing thermal runaway have been reported worldwide in electronics devices such as cell phones, laptops, and electric vehicles, and even auxiliary power units powering commercial aircrafts [6]. Although these incidents are highly unlikely, they are reminders that safety is a prerequisite for batteries, whatever the level of innovation, and that understanding the causes and processes of thermal runaway of high-energy LIBs before their applications is essential to guiding the design of functional materials and improving the safety and reliability of LIBs.

Battery safety is becoming even more critical with the emergence of highly reactive Ni-rich LIBs in the market. These batteries are commercialized to meet novel energy- or power-demanding applications and are expected to dominate the market in the coming years, likely until the occurrence of a new technological breakthrough. This novel battery generation of such high energy density and more intrinsically reactive materials could possibly lead to more catastrophic events involving thermal runaway. Future safe and sustainable use of such innovative chemistry batteries requires at an early stage a comprehensive characterization of the properties impacting their safety profile. In this context, there is a clear need to better understand the underlying specific electrochemical and thermal behaviors of these cells in both normal and abuse conditions across their lifetime. This is a prerequisite for adapting experimental, analytical, and modeling tools, as developed in previously performed studies, which are essentially validated for more mature and less reactive positive and negative chemistry combinations [7–9] in order to expand their applicability to innovative energetic chemistries.

Accordingly, and inspired by the works of Abada et al. [10–12], Panchal et al. [13–15], and other researchers [16,17], this research aims to go deeper into the understanding of this complex multiphysics phenomenon, featuring the thermal runway process at cell scale, taking into account the intertwined impact of SOC level and the relating properties of highly reactive Ni-rich technologies, such as electrode materials (NMC811 or NCA positive electrodes in combination with graphite-$SiO_x$ composite negative electrodes), separators, and the thermal propagation of cell core and safety features.

In this article, the methodology for investigating the thermal runaway of pristine cells through experimental study is presented. First, the results confirmed the proposed complete thermal runaway exothermic chain reactions for high-energy, Ni-rich LIBs. Then, the different factors impacting the thermal runaway kinetics are addressed. Finally, the relationship between safety features and SOC towards venting and component ejection mechanisms are demonstrated.

As the critical first step for the future research on the safety of LIBs, this experimental work led us to a clearer understanding of the thermal runaway influenced by both cell technology and SOC. The detailed findings will help to adjust and upgrade the multiphysics modeling tool as needed under COMSOL, which so far has been calibrated for Lithium iron phosphate $LiFePO_4$ (LFP) cells developed by [10].

## 2. Experimental Methodology of Investigating the Thermal Runaway

The experimental methodology includes:

- The complete multiscale cell analysis, in order to address the pristine and thermally abused states of LIBs;
- The thermal abuse tests, in order to perform and investigate the thermal runaway phenomenon.

These interconnected experimental processes are illustrated in Figure 1.

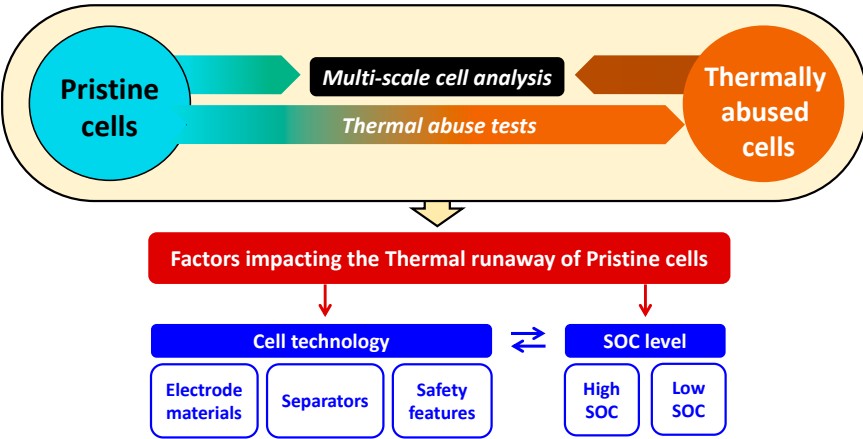

**Figure 1.** Research experimental methodology.

The thermal abuse tests consider the intertwined impact between cell technology and SOC level. The complete multiscale cell analysis supports the interpretation of the multiscale phenomena, ranging from internal physico-chemical characterization to battery components reactions (electrodes, electrolytes, and separator), and further to the thermal propagation of cell core and safety features (pressure disk, button vent, gasket seal, open center core, etc.) involved in the thermal runaway process.

### 2.1. Multiscale Cell Analysis

The cells were analyzed at two states: in pristine condition and after thermal runaway occurrence. The complete multiscale analysis of LIB cells is illustrated in Figure 2. The following list outlines the electrical analysis benefits from the common LIB terminologies, namely SOC and C-rate:

- SOC represents the rate of the available capacity to the maximum capacity when a battery is completely charged [18];
- Charge/discharge C-rate is the measurement of the rate at which a battery is charged/discharged relative to its maximum capacity (e.g., C/20 C-rate means that the current will charge/discharge the entire battery in 20 h ideally, therefore, for a battery with a capacity of 3Ah, this is equivalent to a charge/discharge current of 0.15A).

At the cell scale, electrical analysis at 25 °C firstly activates the pristine battery (by 3 standard cycles to form the solid electrolyte interphase (SEI) layer), and secondly provides information on the cells' state of health (such as actual capacity by C/20 test, impedance distribution by galvano electrochemical impedance spectroscopy (GEIS) and pulse power characterization (PPC), kinetic and transport properties of battery electrodes by galvanostatic intermittent titration technique (GITT)) and cell performance (rate capability by cycling test at different charge/discharge C-rates). X-ray tomography is the preferable analysis method to investigate the internal structure of a battery, especially for cell safety features that directly link to the venting mechanism and for battery types with a central core metallic structure.

At the component scale, scanning electron microscopy (SEM) and energy-dispersive X-ray spectroscopy (EDS) methods can give knowledge about the morphology and geometry of cell components, especially about the cell electrode active grains and different layers of the separator.

At the material scale, chemical mapping in combination with X-ray diffraction (XRD) analysis can be used to indicate the existing chemistry or the appearance of chemical processes upon cycling in the cell electrodes, and to identify the active material stoichiometry in the electrode grains. The electrolyte can be analyzed by gas chromatography-mass spectrometry (GC-MS) method.

Additionally, techniques based on differential scanning calorimetry (DSC) measurements should be implemented to study the thermal stability of the materials and components used in selected LIBs (electrode materials, electrolytes, separators, etc.) and the degradation products.

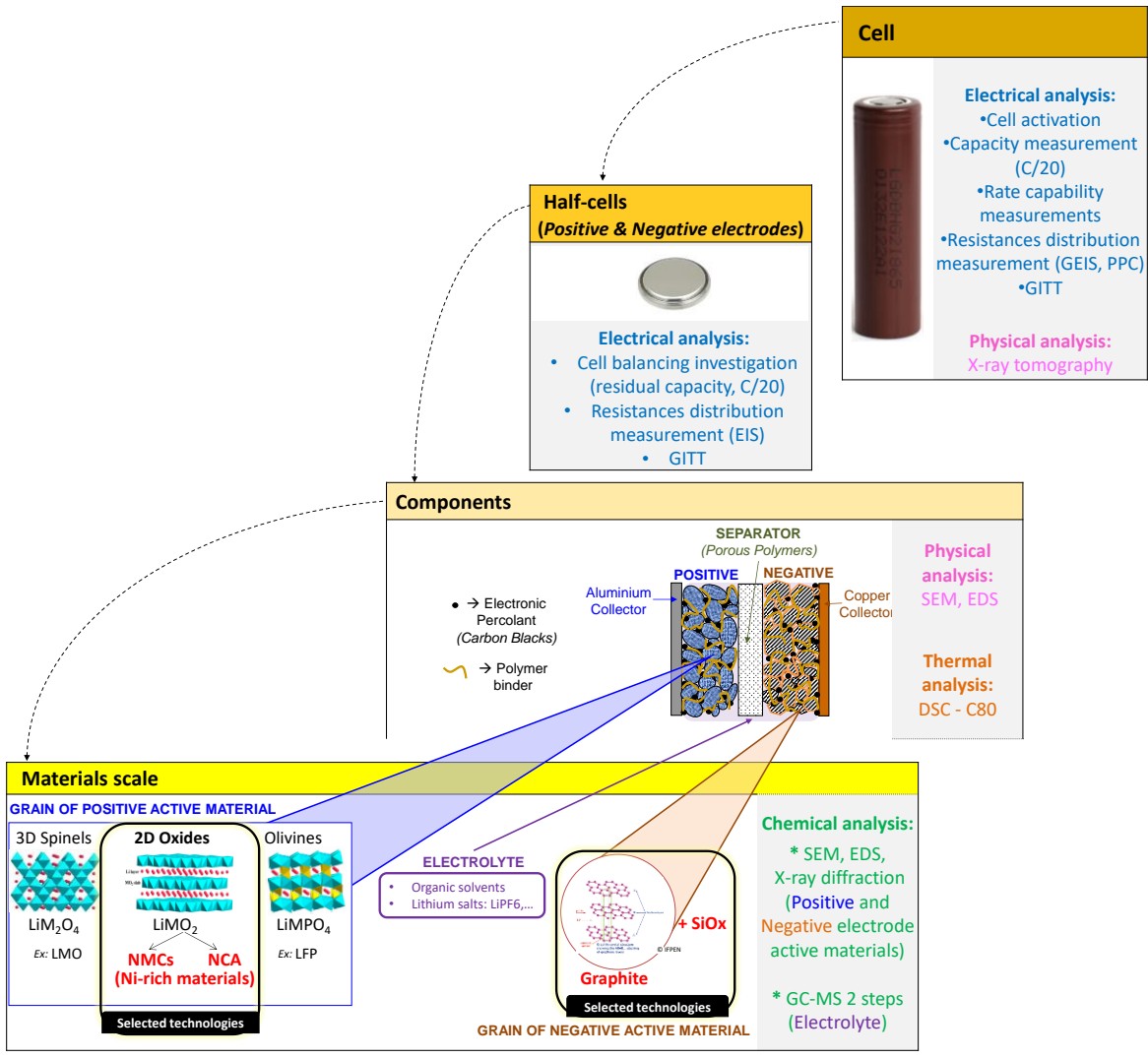

**Figure 2.** Process of the complete multiscale analysis of Lithium-ion battery cell.

### 2.2. Thermal Abuse Tests

Thermal abuse is the most direct way to exceed the thermal stability limits of a lithium-ion battery. Therefore, thermal abuse tests will be carried out to subject the cell to external heating. Pristine cells, in known and quantified states of health as assessed by electrical analysis (Figure 3), are currently progressively undergoing thermal abuse tests on the STEEVE platform at INERIS. These tests make it possible to understand the processes involved in the thermal runaway of the batteries.

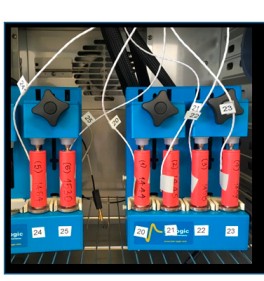 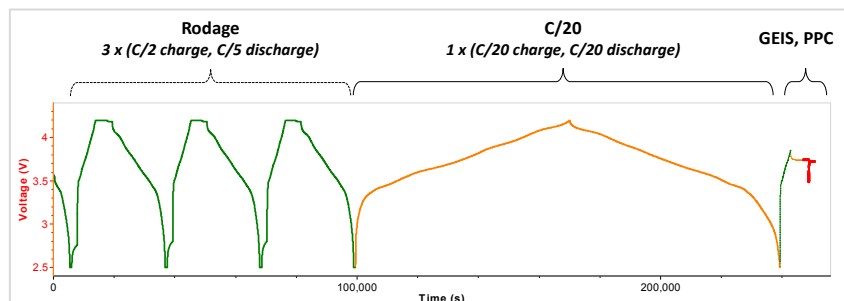

**Figure 3.** Setup of pristine cells in 25 °C chamber (**left**) and electrical analysis protocol (**right**).

Studying the thermal runaway process requires not only elevated temperature, but also an adiabatic (highly insulated) environment and extended time to reach a self-sustaining thermal runaway condition. The quasi-adiabatic conditions in accelerating rate calorimetry (ARC) tests can be regarded as perfect thermal insulation conditions. Therefore, inspired by [10], the results of ARC tests represent a worst case scenario, where the safety behavior during the thermal runaway of the battery cell is mainly characterized by the three critical thresholds below:

- $T_1$ is the temperature corresponding to the initial detected self-heating. The detection of $T_1$ depends on the detection sensitivity of the testing device for the exothermic reaction, which is indicated in the testing program. In this work, the detection sensitivity for exothermic reactions is ≥0.03 °C/min;
- $T_2$ is the temperature referred to the initial observation of the endothermic reaction of the separator melting process;
- $T_3$ is the temperature corresponding to the sudden acceleration of temperature rate due to the final venting, indicated by the strongest gassing rate and the hard ISC after the ceramic layer has collapsed.

These critical temperatures can be identified based on the evolution of cell skin temperature and confirmed by the behaviors of cell self-heating temperature rate ($T_{rate}$) during thermal runaway, where:

$$T_{rate} = dT/dt \tag{1}$$

Another important parameter is the pressure rate ($P_{rate}$) of BTC's vessel, which indicates the rate of venting events, gas ignition, fire, and chemical explosion, where:

$$P_{rate} = dP/dt \tag{2}$$

ARC tests are typically operated by the heat–wait–search (HWS) algorithm [19]. In such experiments, the heat is not allowed to be transferred from the cell to its surroundings. The applied HWS test protocol is illustrated in Figure 4.

The system is firstly stabilized at a certain initial temperature (35 °C in this case), then the battery cell is heated at increments of 5 °C, while the system maintains the adiabatic conditions. After wait and search periods, if the self-heating rate is not significant ($T_{rate} < 0.03$ °C/min), the HWS loop will be resumed. After a significant exothermic reaction is detected, the ARC changes into the exothermic tracking mode, where it follows the cell temperature adiabatically. At this point, the test would come back to the HWS loop if the $T_{rate} < 0.01$ °C/min. After 450 °C, the exothermic tracking mode continues until the end of thermal runaway.

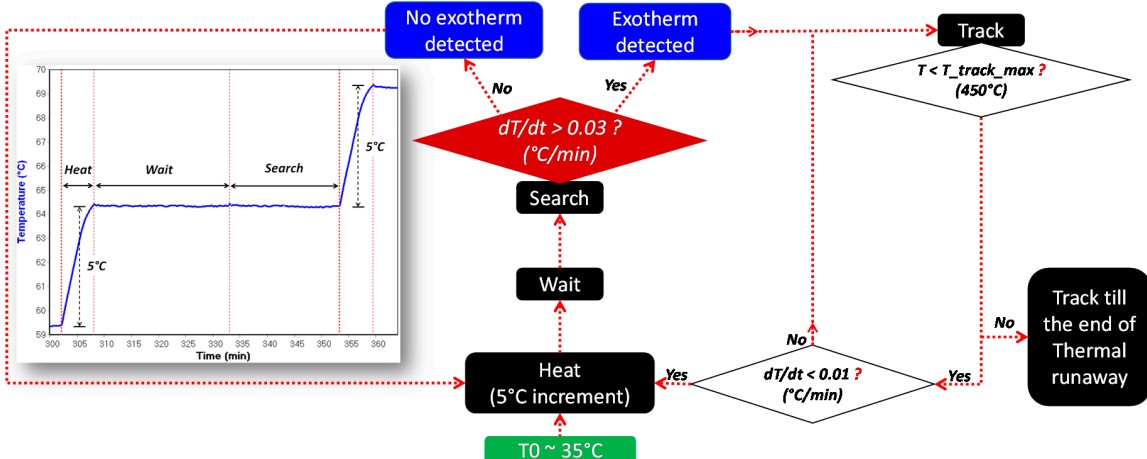

**Figure 4.** The applied HWS test protocol in quasi-adiabatic conditions.

## 2.3. Technology Selection

With the quickly growing LIB market, the supply of Ni-rich LIB technologies today presents less difficulties than before. The size (in term of capacity) and format of a cell can have a significant effect on safety behavior [6]. For simplification, the cell format studied in this research is fixed as the cylindrical 18650 cell format because:

- The cylindrical format is one of the most common cell formats and the basic phenomenology of the provided battery chemistry is the same, while difficulties may arise from mechanical aspects;
- 18650 size: 18650 cells (e.g., from Japanese/Korean manufacturers) have been widely used by consumers and electric vehicles for quite a while, and according to international regulations for transport of dangerous goods (UN TDG model regulations), these are subject to reportable quality control procedures. Therefore, functional and safety performance repeatability could be assumed as representative of best products currently available on the market;
- The detection of a temperature rise could be easier in the case of small cells, since only one or two sensors (thermocouples) might be needed for detection. In contrast, for larger cells, more sensors are needed. The thermocouple type used in this research is the K-Type Inconel 600 Class 1, which has accuracy of ±1.5 °C from −40 °C to 375 °C and ±0.004 × T (°C) from 375 °C to 1000 °C.

Two 18650 Ni-rich, high-energy batteries from the new Li-ion battery generation were selected, namely a LG 18650 HG2 and a Panasonic NCR 18650 GA. A complete analysis was performed on pristine cells in order to carefully check the cell chemistry, and thereby to confirm the choice of highly reactive Ni-rich LIBs technologies studied. The LG HG2 and Panasonic NCR GA are based on NMC811 and NCA, respectively, as the positive electrode active materials, and with graphite-SiO$_x$ composite technologies as the negative electrode active materials. More details are found in Table 1.

**Table 1.** Selected Ni-rich commercial batteries.

| Cell Technology | LG 18650 HG2 | Panasonic NCR 18650 GA |
|---|---|---|
| Cell chemistry (Identified by multiscale cell analysis in Section 2.1.): | NMC811/(graphite-SiOx) | NCA/(graphite-SiOx) |
| Electrolyte | $LiPF_6$ in EC-DMC-PC [1] | $LiPF_6$ in EC-DMC-DEC-PC [1] |
| Format | cylindrical | cylindrical |
| Diameter × Length | 18 mm × 65 mm | 18 mm × 65 mm |
| Weight | 44.56 g ± 0.2 g | 47.26 g ± 0.2 g |
| Nominal capacity | 3000 mAh | 3450 mAh |
| Charging voltage | 4.20 V ± 0.05 V | 4.20 V ± 0.03 V |
| Cut off voltage | 2.5 V | 2.5 V |
| Standard charge | 1500 mA (C/2) | 1725 mA (C/2) |
| Standard discharge | 600 mA (C/5) | 690 mA (C/5) |
| Operating temperatures (from manufacturer) | Charge: 0 °C~50 °C Discharge: −20 °C~75 °C | Charge: +10~+45 °C Discharge: −20~+60 °C |

[1] Under further investigation. $LiPF_6$ = lithium hexafluorophosphate; EC = ethylene carbonate; DMC = dimethyl carbonate; PC = propylene carbonate; DEC = diethyl carbonate.

Each selected technology shows a very good repeatability, with few cell-to-cell variations. The differential capacity curves electrochemically prove that these technologies are different, as illustrated in Figure 5.

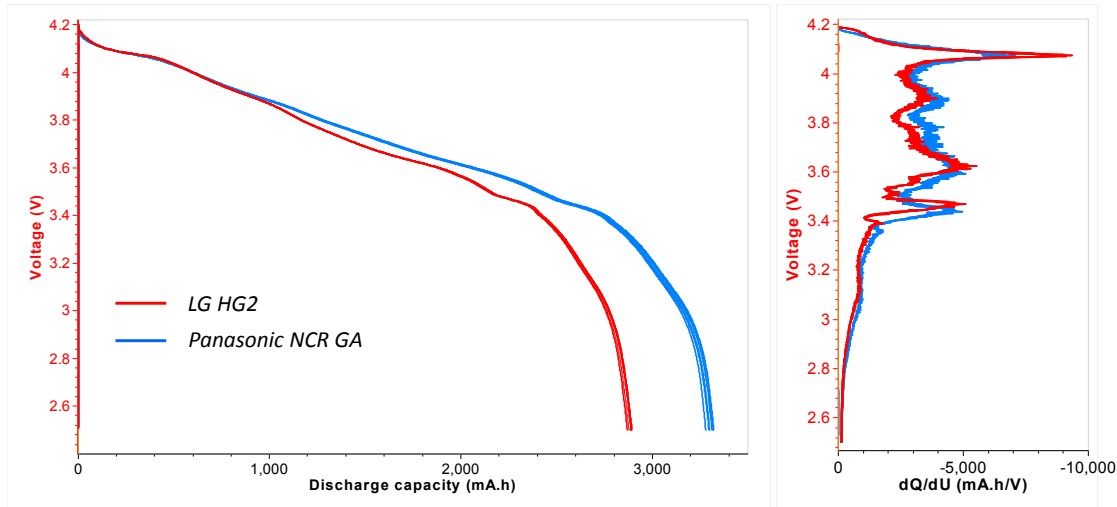

**Figure 5.** The actual discharge capacity profiles (**left**) and the differential capacity curves (voltage vs. dQ/dU) (**right**) of the selected technologies in their pristine states (plot of 6 cells for each technology, obtained from C/20 tests using the Electrochemistry Bio-Logic science instrument and EC-Lab software).

## 3. Results and Discussion

These ARC HWS reproducible results presented below were performed in a BTC500 E1735, with the cell setup shown in Figure 6. An example of a reproducible test is provided in Appendix A.

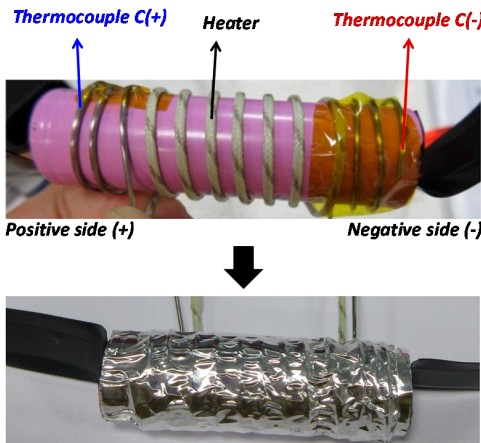

**Figure 6.** Test sensors (thermocouples C (+) and C (−)) and heater setup on 18650 cells.

Each thermocouple is well winded around each end of the cell. Its measuring tip must be in good contact with the cell skin throughout the test, especially during the thermal runaway. These two thermocouples and the heater are secured on the cell skin surface with Kapton tape and then wrapped around one layer of aluminum tape, as seen in Figure 6. This will help to increase the temperature homogeneity on the complete cell skin surface, as well as avoid the impact of different cell plastic skin colors and reduce the heat transfer, since the shiny aluminum surface has low emissivity.

The thermocouple C (+) is near the cell cap (positive side), therefore, its temperature represents the influence of the safety features during venting and combustion. On the other hand, the results of thermocouple C (−) will be more representative of the behaviors and thermal stability of the cell chemistry.

### 3.1. Thermal Runaway Chain Exothermic Reactions of Ni-Rich Lithium-Ion Batteries

Based on our experimental work and inspired by the studies on the abuse behavior of different 18650 cells [3,10–12,20,21], the thermal runaway chain exothermic reactions of Ni-rich LIBs based on negative graphite electrodes can be summarized in Figure 7.

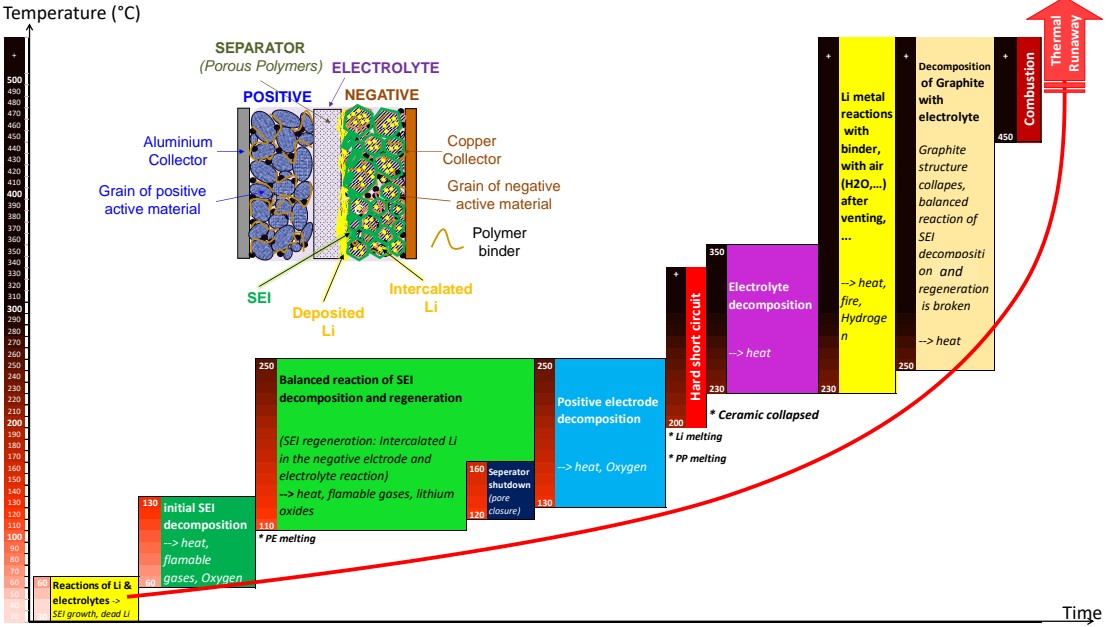

**Figure 7.** Thermal runaway chain exothermic reactions of Ni-rich lithium-ion batteries.

It should be noted that these reactions do not strictly happen one after another in the given order. They are rather complex and systematic issues. The reactions relating to the deposited Li are for the case of Li deposition (Li plating) occurring in the cell.

Inspired by [10], based on the three critical temperatures $T_1$, $T_2$, $T_3$, the thermal runaway exothermic chain reactions can be analyzed in five stages in the evolution of the cell skin temperature, as described below.

Stage 1 corresponds to the safe region, where there is no cell self-heating or the cell self-heating is insignificant ($T_{rate} < 0.03\,°C/min$ in this research), and therefore not detected.

Stage 2 is the first reversible self-heating region, which begins from $T_1$—the initial significant self-heating detected when $T_{rate} \geq 0.03\,°C/min$. Above $T_1$, the battery operation changes to an abnormal state caused by the main reactions listed below:

- Reactions of deposited Li and electrolytes, namely SEI and "dead Li" formation;
- Initial decomposition of SEI (dominant reaction of this stage). This is regarded as the first side reaction during the full thermal runaway process. It occurs around 60–130 °C. The exact temperature range depends upon cell chemistry, the thickness of SEI, and SOC level;
- SEI regeneration: Once the SEI decomposes, the intercalated Li in the graphite negative electrodes can contact the electrolyte again, thereby regenerating SEI.

Stage 3 is defined as the last reversible self-heating region, which starts from $T_2$, the initial observation of the endothermic reaction of the separator melting process. Above this critical temperature, the separator loses its mechanical integrity then starts collapsing, which demonstrates the beginning of cell destruction at the component scale. This reaction depends on the composition and porous structure of different separator layers, as well as the separator thickness. It usually starts with the separator layer that has the lowest melting point. The melting points of commercial separators are about 110–130 °C for the polyethylene (PE) layer, 160–170 °C for the polypropylene (PP) layer, and 180–260 °C for the ceramic coating layer. This leads to the soft internal short circuits (ISCs), and most importantly the decrease of cell temperature. Therefore, the separator is an important element in terms of safety. Its endothermic melting process can be initially observed by the decrease of cell temperature during tracking in adiabatic conditions (e.g., Figure 8c) and can be confirmed by the decrease of $T_{rate}$.

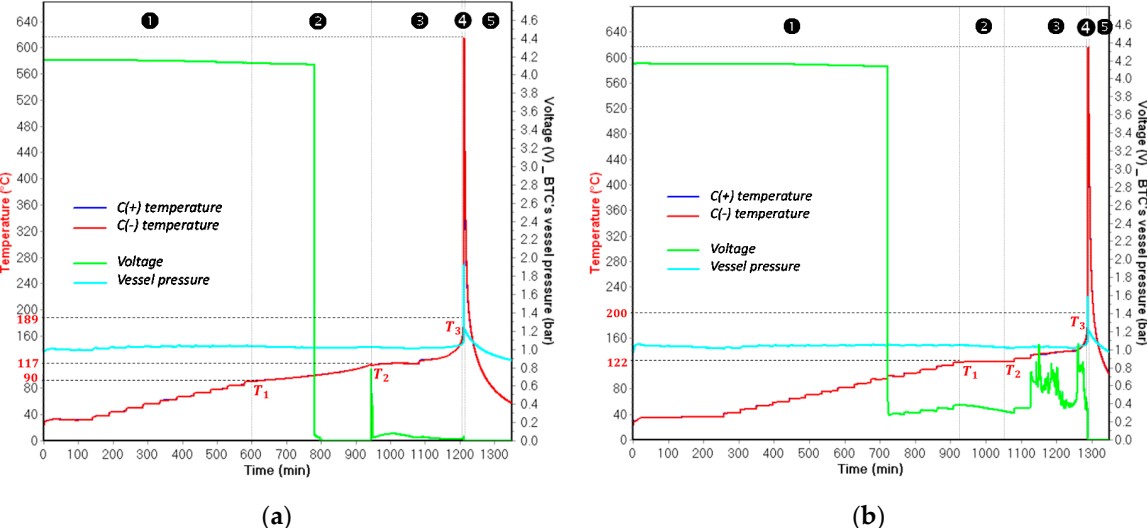

(**a**)  (**b**)

**Figure 8.** *Cont.*

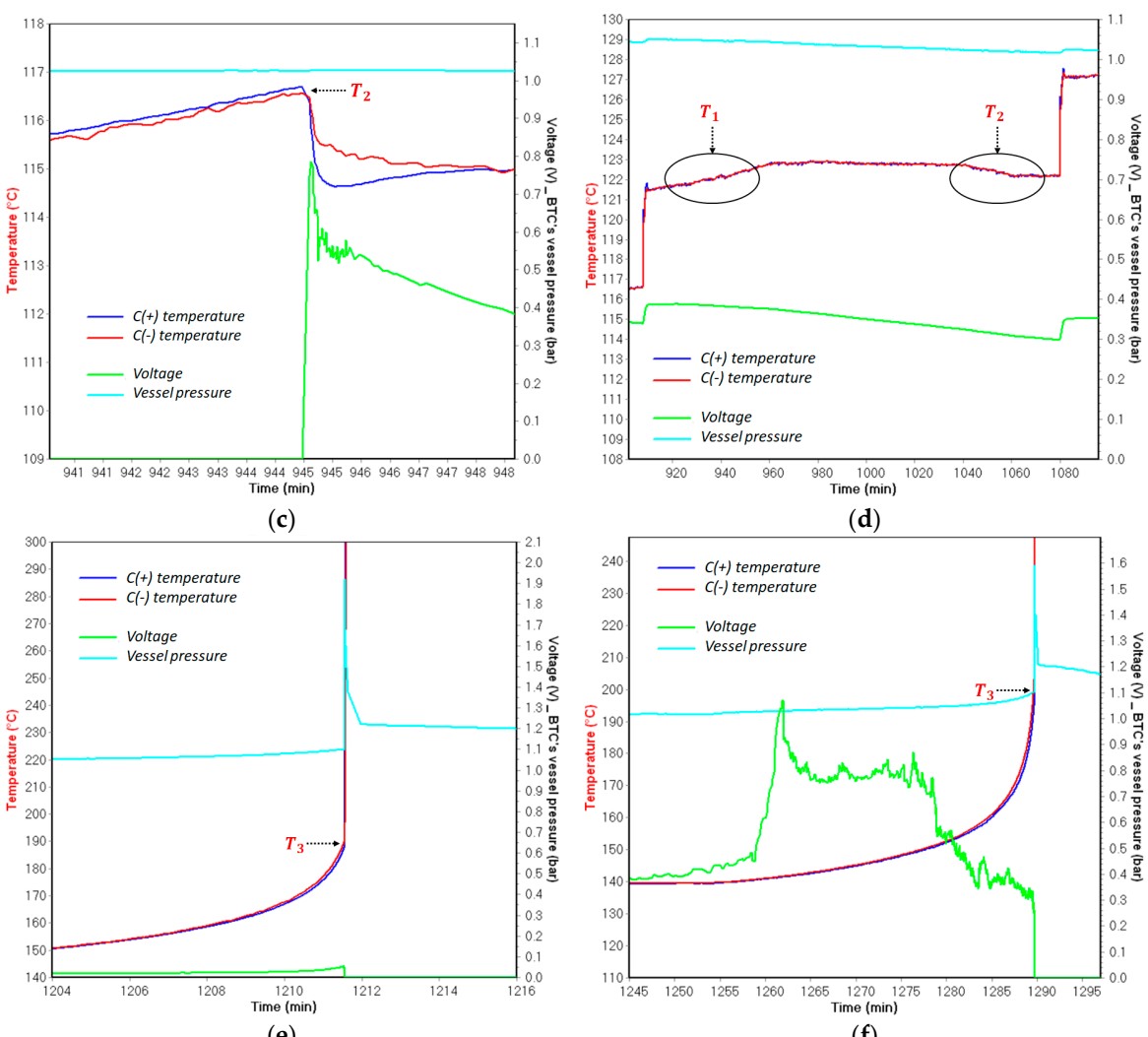

**Figure 8.** (**a**,**b**) The 5 stages in the evolution of cell temperatures obtained from the heat–wait–search (HWS) test of LG HG2 at 100% SOC and of Panasonic NCR GA at 100% SOC, respectively; (**c**,**d**) the initial observations of the endothermic reaction of separator melting of the LG HG2 at 100% SOC and Panasonic NCR GA at 100% SOC, respectively; (**e**,**f**) the final venting and hard ISC after the ceramic separator layer collapse occurs at $T_3$ for both the LG HG2 at 100% SOC and Panasonic NCR GA at 100% SOC.

During stage 3, the temperature violently increases and oxygen accumulates inside the battery due to these exothermic reactions:

- Further electrolyte balance reaction of SEI regeneration and decomposition at negative electrodes (dominant reactions at the beginning of this stage): Within the temperature range of about 120–250 °C, the SEI decomposition will not stop as long as there is sufficient regenerated SEI, meanwhile the SEI regeneration will not increase because the surface of the negative electrode is still covered by a certain thickness of the SEI layer. This exothermic process represents the balance reaction of SEI decomposition and regeneration, with the average thickness of SEI remaining at a stabilized level;
- The start of positive electrode decomposition and electrolyte oxidation (dominant reactions at the end of this stage): The highly exothermic decomposition of positive electrode starts at temperatures ranging 130–200 °C and produces oxygen.

Therefore, stage 3 is also referred to as the heat accumulation and gas release process. During this stage, the cell might vent due to the cell's internal pressure increase caused by the solvent vaporization

and the gases generated. The initial venting events can be observed by the sudden increases of $P_{rate}$ during adiabatic tracking conditions.

The beginning of the final venting is indicated by the strongest gassing rate and the hard ISC after the ceramic layer has collapsed, which simultaneously happen at a similar temperature level ($\sim T_3$) (e.g., Figure 8e). These two events lead to the accelerated heat accumulation and activate the battery combustion as soon as there is enough oxygen (mainly from the positive electrode decomposition reactions and from the air). This is the start of stage 4, the accelerated thermal runaway region, where the cell self-heating is accelerated and irreversible. This begins at $T_3$ and leads to the maximum temperature ($T_{max}$). The internal temperature rate at $T_3$ of these selected Ni-rich technologies in pristine state is ~48 °C/min. During this stage, combustion occurs, vessel pressure aggressively increases, and venting and cell component ejection accelerate, causing fires and chemical explosion hazards due to the strong exothermic reactions below:

- Further highly exothermic decomposition of positive electrodes (~200–250 °C): Cells manufactured from the higher energy density positive electrode materials will be subjected to more severe exothermic reactions;
- Strong exothermic reaction between oxygen (released from positive electrode) and electrolytes;
- Decomposition of electrolytes (combustion reactions): A part of the carbonate electrolyte decomposes inside the cell and releases gases, whereas the other part evaporates and bursts out of the cell. During the final venting, the ultimate severity of the reaction is dominated by the ignition of flammable vent gases. Auto-ignition temperatures of some typical organic electrolyte components are ~440 °C for ethyl methyl carbonate (EMC), ~445 °C for diethyl carbonate (DEC), ~455 °C for propylene carbonate (PC), ~458 °C for dimethyl carbonate (DMC), and ~465 °C for ethylene carbonate (EC) [4];
- Reactions between deposited Li and air ($H_2O$) diffused into the cell after venting;
- Reactions between deposited Li and the binder;
- Decomposition of negative electrodes with electrolytes: The balance reaction of SEI decomposition and regeneration is broken as the temperature increases. Then, the graphite structure collapses;
- Binder reactions.

Stage 5 is the cooling down stage.

The chain exothermic reactions happen under near adiabatic conditions, where the exact reaction temperature highly depends on the cell chemistry and SOC level. Therefore, the temperature rate and duration of these thermal runaway stages are representative of the characteristics of the battery cell.

Soft and hard internal shorting is often the predominant reason for thermal runaway activation, but these factors are relatively hard to control.

*3.2. The Thermal Runaway of Selected Ni-Rich Technologies*

The representative characteristics of the thermal runaway chain exothermic reactions of LG HG2 and Panasonic NCR GA at pristine state of 100% SOC are analyzed in Figure 9.

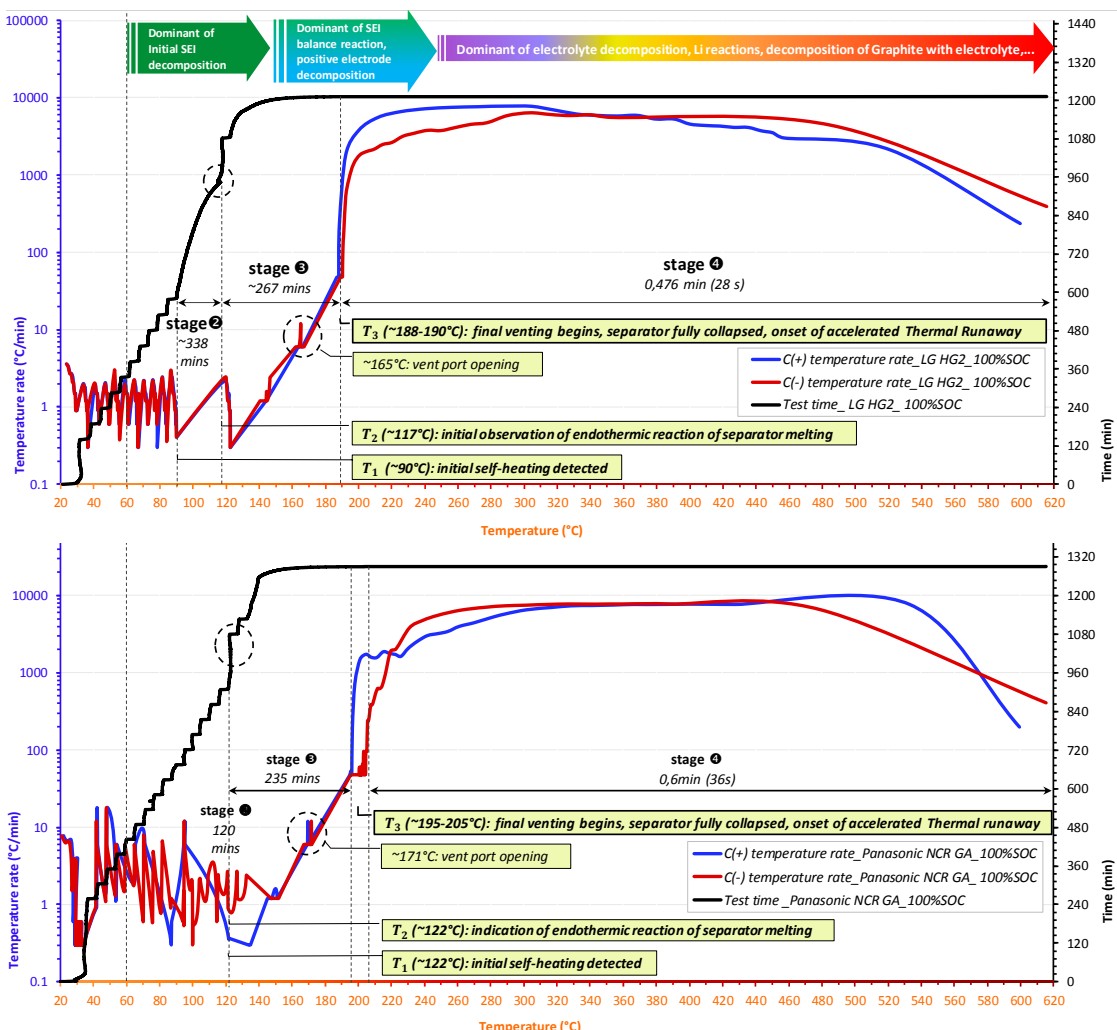

**Figure 9.** Temperature rates and test time versus temperature during the thermal runaway of the LG HG2 (**top**) and Panasonic NCR GA (**bottom**) pristine cells at 100% SOC.

The initial significant self-heating of LG HG2 was detected at a lower temperature (~90 °C) compared to that of the Panasonic NCR GA (~122 °C). This might originally be due to the higher quantity of SEI formed in LG HG2 cells after the electrical analysis (Figure 3), which leads to a stronger initial SEI decomposition/regeneration reaction occurring, since this technology has larger negative electrode active material surface compared to that of Panasonic NCR GA. Consequently, this demonstrates that the safe region (stage 1) of the Panasonic NCR GA is wider than that of LG HG2. Specifically, within the temperature range of 90 °C to 122 °C, the Panasonic NCR GA at 100% SOC is safer than LG HG2 at 100% SOC.

The cell self-heating temperature continues to increase throughout stage 2. The endothermic process of separator melting was initially observed at a similar temperature range for these selected technologies ($T_2$ ~117 °C for LG HG2 and $T_2$ ~122 °C for Panasonic NCR GA), as illustrated in Figure 8c,d. This might be because of the similarities in these separator technologies. It is also seen that in case of Panasonic NCR GA, the first observation of separator melting is at the same temperature (~122 °C) as the first exotherm detected. This means that this technology is safe and that the cell can increase to the temperature of separator melting without any thermal runaway exothermic reactions being detected.

Stage 3 corresponds to the heat accumulation and gas release process, where the self-heating rate of these Ni-rich technologies strongly increases. This is the last reversible self-heating region. It lasts

about 267 min for the LG HG2 and about 235 min for the Panasonic NCR GA. The first venting events that occur during the opening of vent ports were observed at ~165 °C for the LG HG2 and at ~171 °C for the Panasonic NCR GA by the sudden changes of $P_{rate}$ and cell temperature. The final venting and hard ISC of the Panasonic NCR GA occurred at higher temperature (~205 °C) than that of the LG HG2 (~190 °C), as shown in Figure 8e,f.

The severity of the thermal runaway accelerates during stage 4, thereby leading to the maximum temperature of ~615 °C for both technologies. As mentioned in Section 3.1., this intensive stage is caused by the combustion and explosive decomposition, so it lasts only ~28.5 s for LG HG2 and ~36 s for Panasonic NCR GA. During this stage, venting and cell component ejection accelerate (observed in the test videos provided in the Supplementary Materials section).

*3.3. The Factors Impacting the Thermal Runaway of Selected Technologies in Pristine State*

The severity of thermal runaway mainly depends on the cell technology, such as the chemical energy in the form of combustible materials stored in the battery (e.g., high energy density electrode materials and flammable electrolytes), the shutdown mechanism of the separator, and the mechanical design of the safety features.

The severity of the thermal runaway also depends on the SOC level, which indicates the electrical energy is stored in the cell in the form of chemical potential energy.

3.3.1. The Impact of Electrode Materials

The more energy a battery cell stores, the more energetic its thermal runaway will be. Therefore, the thermal runaway reactions of Ni-rich LIBs are very energetic because they have very high energy densities compared to other cell chemistries, especially $LiFePO_4$ (LFP)/graphite. Figure 10 illustrates the severity during the final stages of the thermal runaway for the selected Ni-rich technologies compared to the thermal runaway of a LFP/graphite battery from A123 Systems. A similar type of separator was used, so the hard ISC occurred at similar temperatures (190–210 °C) for all these technologies. It is clearly seen that the Ni-rich LIBs are much more reactive.

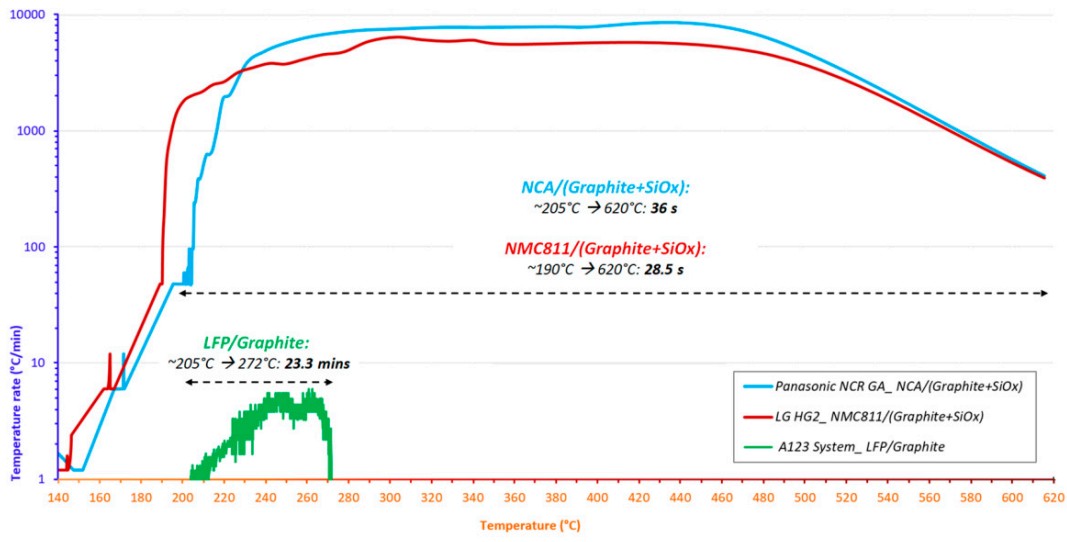

**Figure 10.** Comparison of the final stages of the thermal runaway of the selected Ni-rich technologies versus the "safest technology", $LiFePO_4$ (LFP)/graphite.

Having similar technology to the negative electrode (graphite-$SiO_x$ composites), different positive electrode technologies (NMC811 in LG HG2 and NCA in Panasonic NCR GA) impacted the total duration and the severity of stage 4: With higher temperature rate, the NCA cell lasts ~36 s, while the NMC811 cell lasts 28.5 s, as shown in Figure 9. However, due to the stronger gassing from positive

electrode decomposition of NMC811 technology, the final venting rate detected for LG HG2 is the most violent, as observed in Figure 11 (bottom).

Upon further investigation, we found that the temperature rate during stage 4 of NCA technology reaches a higher value with longer duration, regardless of whether for 100% SOC or 50% SOC. In general, NCA technology appears to be more reactive during the final stage of the thermal runaway than that of NMC811 technology.

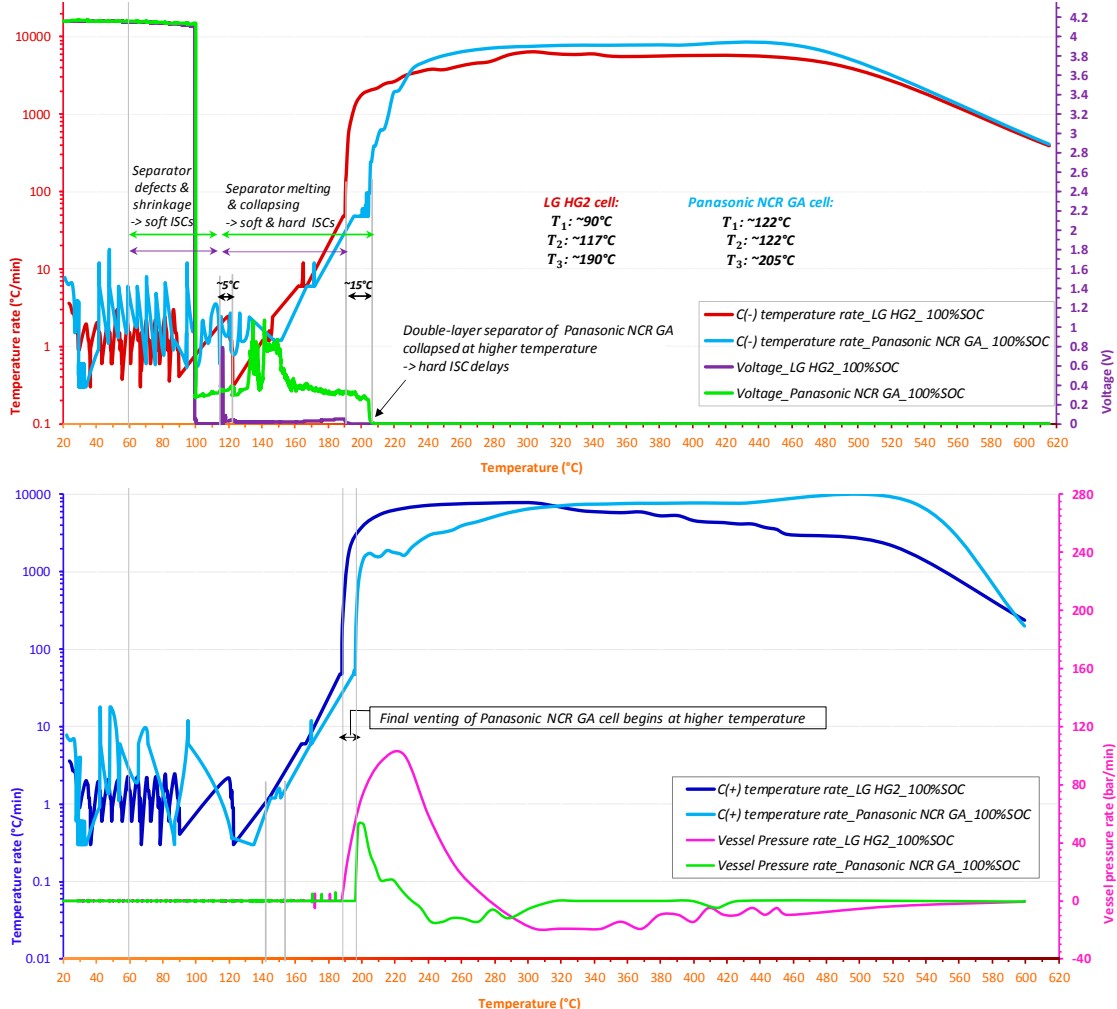

**Figure 11.** The C (−) temperature rate and cell voltage of LG HG2 and Panasonic NCR GA (at 100% SOC) versus C (−) temperature (**top**). Temperature rate of C (+) and vessel pressure rate of LG HG2 and Panasonic NCR GA (at 100% SOC) versus C (+) temperature (**bottom**).

### 3.3.2. The Impact of Separator

As presented in Figure 12, the two selected technologies have similar double-layer separator technology: polymeric and ceramic layers.

For LG HG2, the polymeric layer has homogeneous porosity and the ceramic layer is mainly composed of grains. The thickness ration of the ceramic layer over the polymeric layer is ~1/12 and the separator total thickness is ~13 μm.

For Panasonic NCR GA, the separator polymeric layer has inhomogeneous porosity. The ceramic layer is composed of long fibers in a mixture with different grains. The thickness ration of the ceramic layer over the polymeric layer is ~1/2. The separator total thickness is ~38 μm, which is significantly thicker than the separator for LG HG2. Therefore, the Panasonic NCR GA's separator fully collapses at higher temperature, which leads to the delay of final venting, as shown in Figure 11 (bottom), and also

to the delay of hard ISC (~205 °C compared to ~190 °C for LG HG2), as observed in Figure 11 (top). Therefore, stage 4 of Panasonic NCR GA starts at a higher temperature.

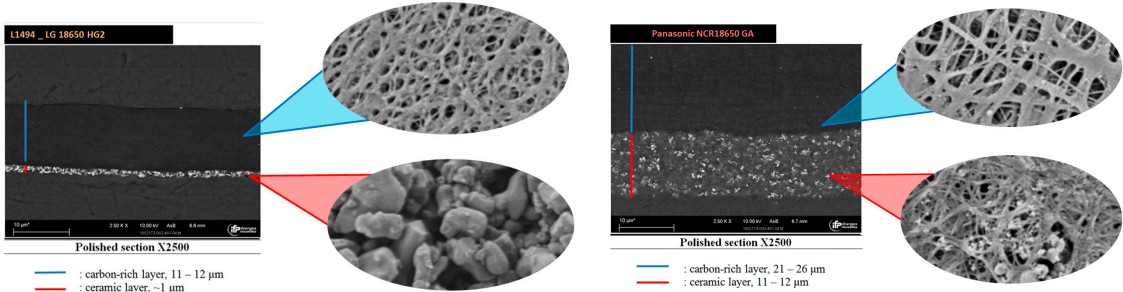

**Figure 12.** SEM images of LG HG2 (**left**) and Panasonic NCR GA (**right**) double-layer separators (results from the Physical Research and Analysis division of IFPEN, 2018).

### 3.3.3. The Impact of SOC

For any given cell, the most severe thermal runaway process is achieved when that cell contains its maximum electrical energy (100% SOC or overcharged) [17]. To better understand the impact of SOC level on the thermal runaway of LG HG2 and Panasonic NCR GA, these cells were subjected to thermal safety tests at two different levels of SOC: 100% and 50%.

The impact of SOC on the thermal runaway can be observed in Figures 13 and 14. Through testing of multiple batteries, we confirmed that lower SOC level leads to ISC delay, and thereby shifts the initial self-heating as well as the activation of stage 4 to higher temperatures, as shown in Figure 13. Therefore, $T_1$, $T_2$, and $T_3$ are increased for cells at reduced SOC. Additionally, as similarly proven in Figure 14, cells at 50% SOC require more time before undergoing thermal runaway, and their thermal runaway processes can only be activated at higher temperature. Moreover, their stage 4 is less severe, showing a lower temperature rate and shorter duration. Therefore, the pristine cells at 50% SOC are less reactive.

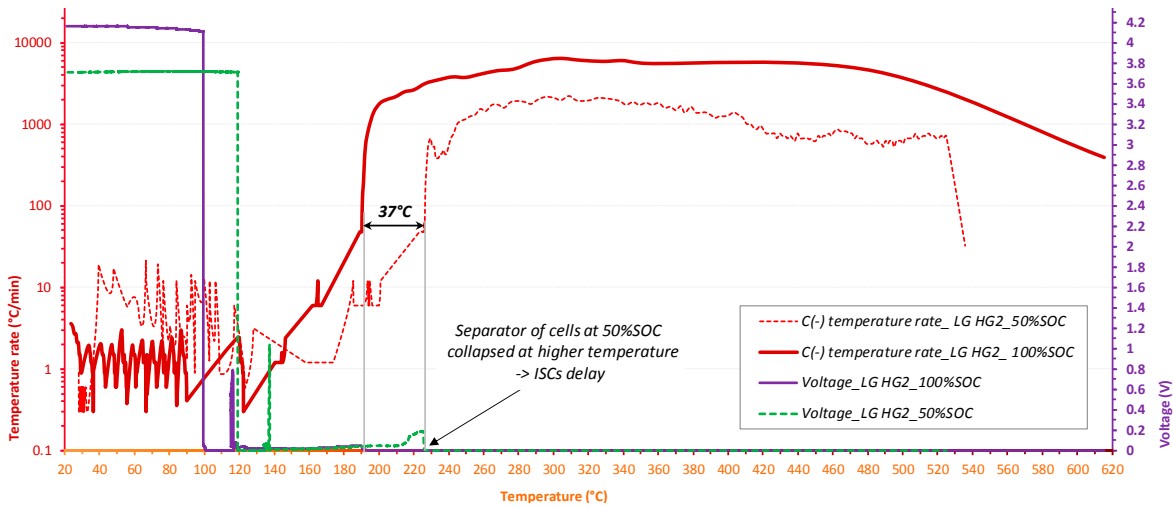

**Figure 13.** The C (−) temperature rate and voltage of LG HG2 at 50% and 100% SOC versus C (−) temperature.

This SOC dependency shift is stronger for the LG HG2 technology, and the maximum temperature is also reduced with reduced SOC (~615 °C for LG HG2 100% SOC and ~535 °C for LG HG2 50% SOC). However, all tested cells clearly exhibit stage 3 and stage 4, with the same temperature rate of ~6 °C/min at the beginning of vent port opening and ~48 °C/min at $T_3$, regardless of SOC and cell technology (Figure 14b).

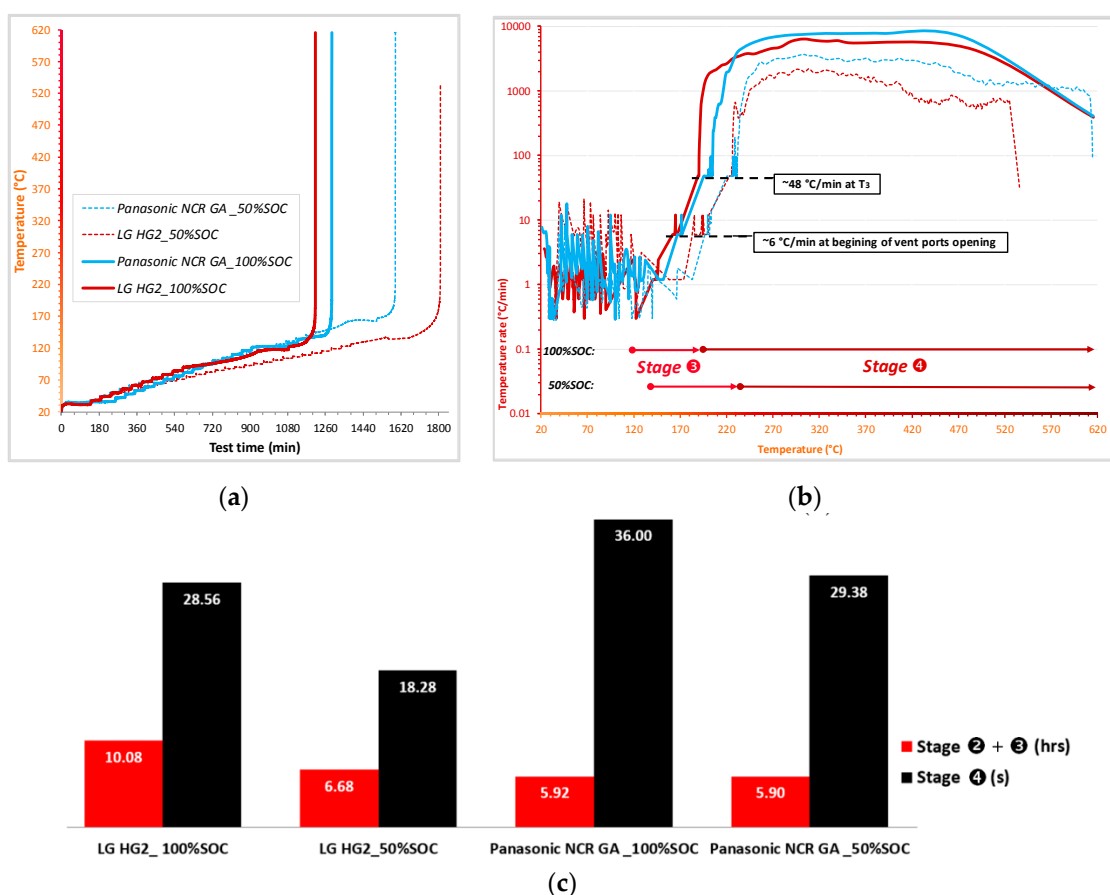

**Figure 14.** (**a**) Temperature profiles versus test time and (**b**) temperature rate versus temperature of the 50% and 100% SOC LG HG2 and Panasonic NCR GA cells. (**c**) Duration of the reversible self-heating region (stage ❷ + ❸) and duration of the irreversible self-heating region (stage ❹) during thermal runaway of the presented cells.

### 3.3.4. The Impact of Safety Features and SOC on Venting and Component Ejection Mechanism

During the thermal runaway process, venting events lead to gas release and cell component ejection. The remaining cells were weighed after having undergone thermal runaway (residuals). As shown in Figure 15, the mass loss of pristine cells at 50% SOC is lower than that of pristine cells at 100% SOC. This additionally confirms the lower reactivity of cells with reduced SOC.

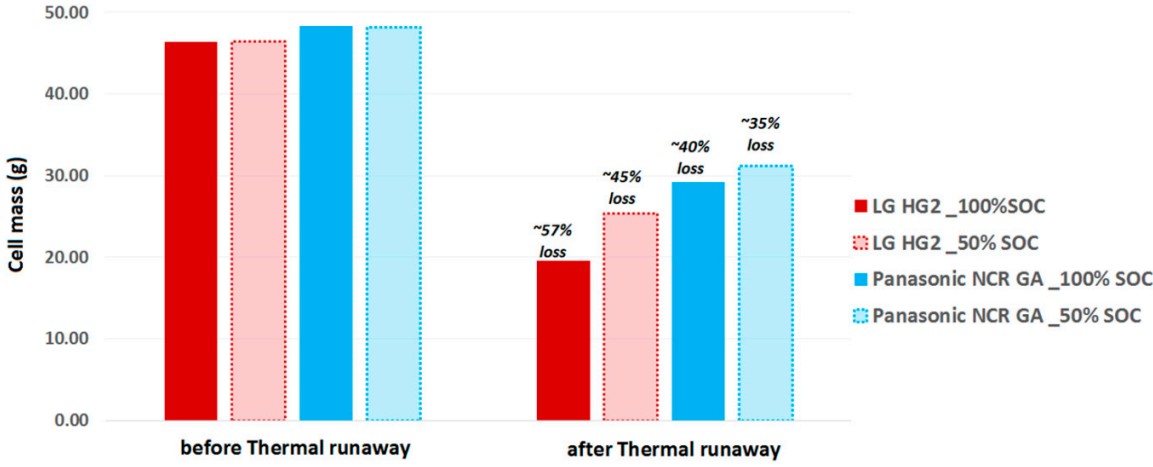

**Figure 15.** Cell mass before and after thermal runaway.

Looking deeper into the impact of SOC on the venting mechanism, although these cells have different safety feature designs, we observed in Figure 16 that during final venting of all cells, the vent ports fully opened; however, the gasket seal collapsed only in the case of cells with 50% SOC. This is also confirmed by the remaining cells after the thermal runaway (presented in Table 2 and Appendix B). This could be explained by the delay of hard ISC in case of 50% SOC, as well as the shift of final venting to higher temperature, where stronger exothermic reactions occur with higher reaction rates, and eventually the pressure rate acceleration during stage 4 collapses the gasket seal. Therefore, the jelly roll of 50% SOC cells was violently ejected (Table 2). This also explains why the duration of stage 4 is shorter for 50% SOC.

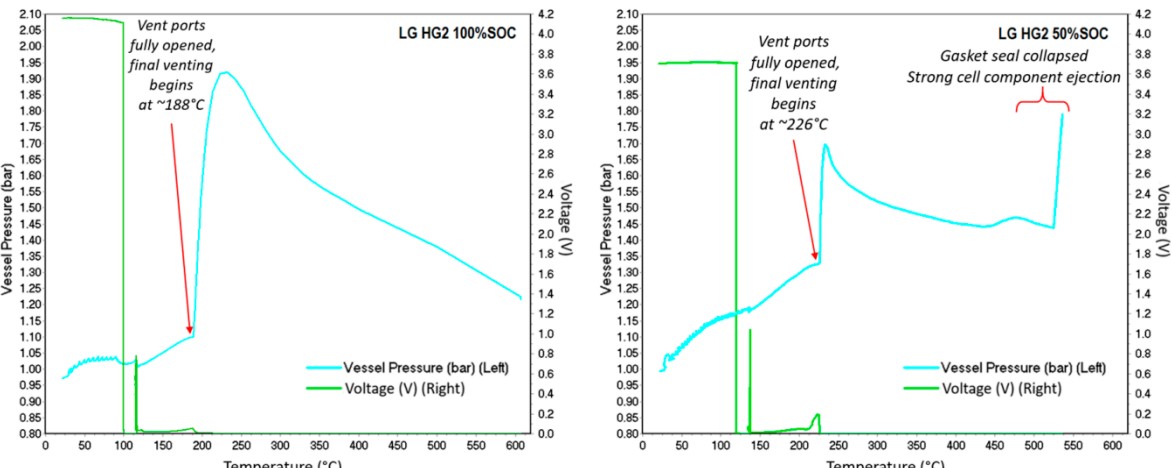

**Figure 16.** Cell voltage and vessel pressure (bar) versus temperature for LG HG2 100% SOC and 50% SOC.

**Table 2.** The impact of safety features and SOC on venting and component ejection mechanisms, as seen through the pictures of LG HG2 and Panasonic NCR GA (50% and 100% SOC) after thermal runaway.

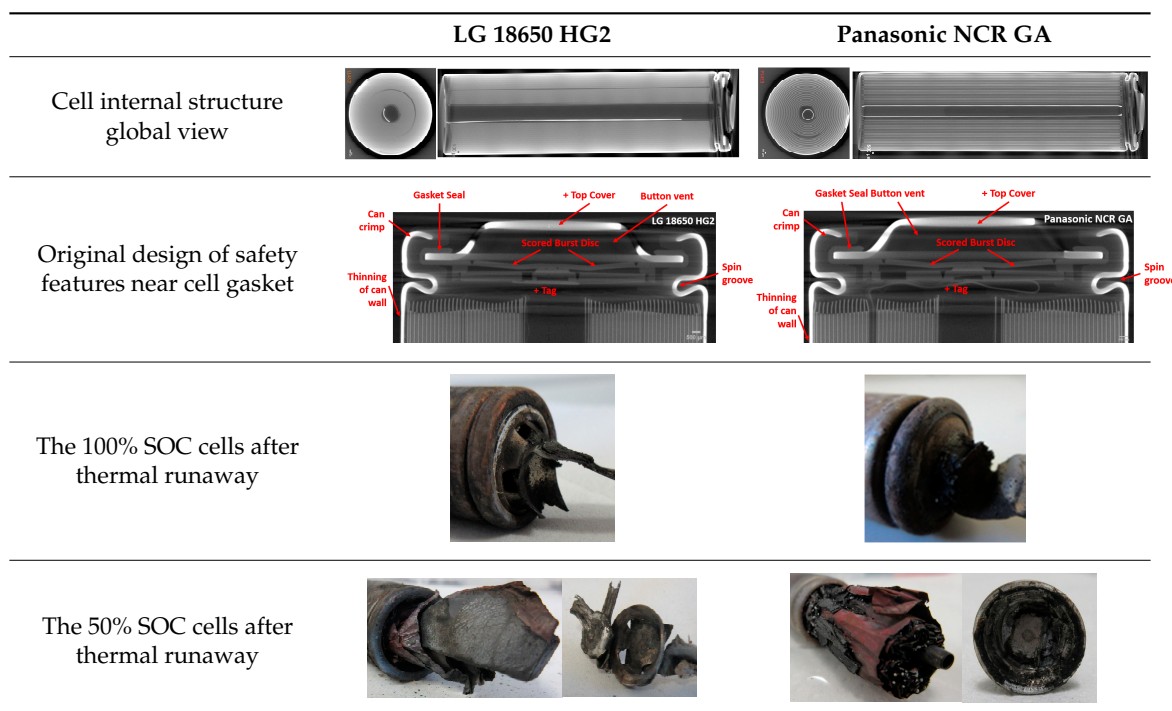

|  | LG 18650 HG2 | Panasonic NCR GA |
|---|---|---|
| Cell internal structure global view |  |  |
| Original design of safety features near cell gasket |  |  |
| The 100% SOC cells after thermal runaway |  |  |
| The 50% SOC cells after thermal runaway |  |  |

The presence of a stiff center tube and a metal bar to allow pressure equalization can be clearly observed in Table 2 in the design of the Panasonic cells and LG cells, respectively (the open center core), preventing winding ejection during thermal runaway. However, these approaches do not work effectively for cells at reduced SOC.

## 4. Conclusions

We performed the thermal abuse tests on the two selected Ni-rich LIBs charged to different levels of SOC (100% and 50%) at pristine states in quasi-adiabatic condition (ARC). The obtained results first confirmed the proposed complete thermal runaway exothermic chain reactions, then discovered the different factors impacting the thermal runaway kinetics, and furthermore demonstrated the relationship between safety features and SOC with venting and component ejection mechanisms. The main findings are below:

(1) For all tested SOC levels, the initial significant self-heating of Panasonic NCR GA was always detected at a higher temperature. Therefore, the safe region of this technology is wider than that of LG HG2. This might originally be due to the stronger initial SEI decomposition/regeneration reaction occurring in LG HG2 cells.

(2) As a critical element in terms of safety, the separator technology significantly impacts the reversible self-heating stages and the onset of the accelerated thermal runaway. Having a 3-fold thicker separator, Panasonic NCR GA exhibited the final venting and hard ISC at the higher temperatures which were ~205 °C for 100% SOC compared to ~190 °C for the LG HG2, and ~228 °C for 50% SOC compared to ~222 °C for the LG HG2.

(3) Regardless of SOC, the positive electrode material strongly influences the severity during the final stage of thermal runaway. NCA technology appears to be more reactive, with a higher self-heating temperature rate over a longer duration (Figure 14c). However, the final venting of LG HG2 is the most violent due to the stronger gassing from positive electrode decomposition due to NMC811 technology compared to NCA technology.

(4) The impact of SOC on the thermal runaway was significantly observed for all selected Ni-rich technologies. We found that the cells at reduced SOC were less reactive. They required more time before undergoing thermal runaway and their accelerated thermal runaway stage had lower temperature rates over a shorter duration and were only activated at higher temperature, thereby meaning their gasket seal collapsed and they exhibited cell jellyroll ejection but lower mass loss after the thermal runaway.

(5) All tested cells exhibited the same self-heating temperature rate of ~6 °C/min at the beginning of vent port opening and ~48 °C/min at the onset of the accelerated thermal runaway ($T_3$).

The notable impacts of aging (SEI-driven aging and Li plating) on the behavior of Li-ion cells in thermal abuse conditions [3,22–25] will be investigated in the next step of this research. Future work will also deal with the calibration and validation purposes of the development of a consolidated 3D thermal runaway model in order to predict the behaviors of different LIBs, at pristine and aged states, near to and during thermal runaway. This thermal runaway model will eventually be transposed into tools enabling the best design of the packs and avoidance of this undesirable phenomenon.

**Supplementary Materials:** The videos of the final stage of the thermal runaway of LG HG2 100% SOC, LG HG2 50% SOC, Panasonic NCR GA 100% SOC are respectively available online at: https://youtu.be/w8OkGQIu6PM, https://youtu.be/rQNZ0Ti8E1A, https://youtu.be/3MbRVwj8w2Q.

**Author Contributions:** Conceptualization, J.B., M.P., S.A., A.L., and T.T.D.N.; methodology, J.B., T.T.D.N., and A.L.; validation, J.B., G.M., S.L., and S.G.; formal analysis, T.T.D.N.; investigation, T.T.D.N., S.A., and A.L.; resources, J.B. and A.L.; data curation, T.T.D.N. and S.A.; writing—original draft preparation, T.T.D.N.; writing—review and editing, all authors; visualization, T.T.D.N.; supervision, J.B., G.M., and S.L.; project administration, J.B. and S.L.; funding acquisition, J.B.

**Funding:** This research received no external funding.

**Acknowledgments:** The authors gratefully acknowledge the support and guidance of IFPEN–INERIS–LRCS in this work, especially the technical support from Electrochemistry and Materials department of IFPEN and Polygon 4 (STEEVE platform) of INERIS.

**Conflicts of Interest:** The authors declare no conflict of interest.

## Abbreviations

| | |
|---|---|
| LIB | Lithium-ion battery |
| SOC | state of charge |
| NMC811 | $Li(Ni_{0.8}Mn_{0.1}Co_{0.1})O_2$ |
| NCA | $Li(Ni_{0.8}Co_{0.15}Al_{0.05})O_2$ |
| 3D | 3 dimensions |
| LFP | $LiFePO_4$ |
| SEI | solid electrolyte interphase |
| GEIS | galvano electrochemical impedance spectroscopy |
| SEM | scanning electron microscopy |
| EDS | energy-dispersive X-ray spectroscopy |
| XRD | X-ray diffraction |
| DSC | differential scanning calorimetry |
| ISC | internal short circuit |
| ARC | accelerating rate calorimetry |
| HWS | heat–wait–search |
| PPC | pulse power characterization |
| GITT | galvanostatic intermittent titration technique |
| GC-MS | gas chromatography-mass spectrometry |

## Nomenclature

| Nomenclature | Description |
|---|---|
| T | cell skin temperature |
| U | cell voltage |
| Q | cell capacity |
| P | pressure of BTC's vessel |

## Appendix A

All ARC tests at pristine state 100% SOC are reproduced at least twice. The obtained results were coherent for all cells, especially for stage 3 and stage 4. An example of a reproducible test for the LG HG2 at 100% SOC is found in Figure A1.

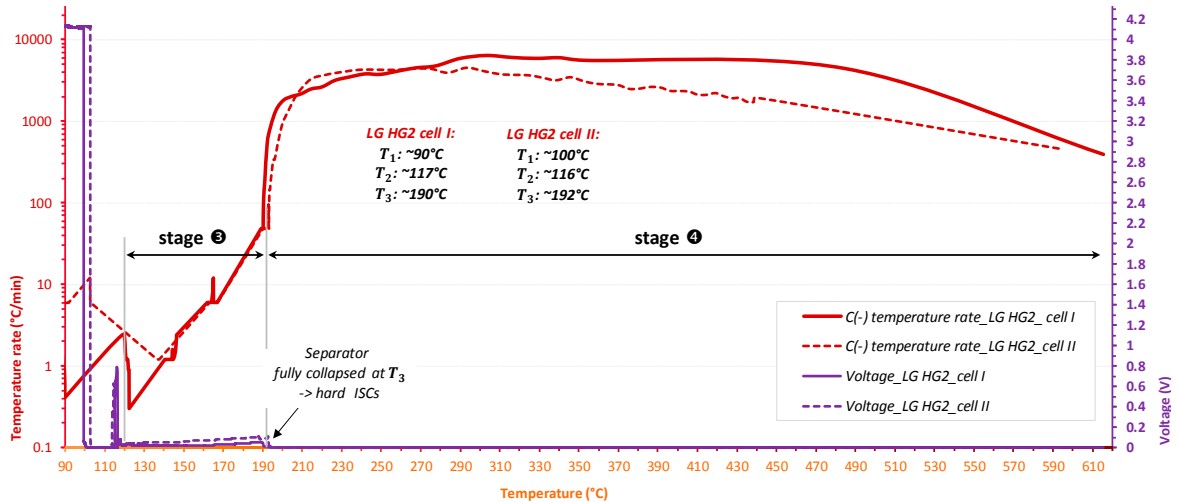

**Figure A1.** Reproducible HWS test results of 2 LG HG2 pristine cells at 100% SOC.

## Appendix B

This appendix presents the supplemental photos of thermally abused cells after undergoing thermal runaway.

**Table A1.** Thermally abused LG HG2 at 100% SOC.

| | |
|---|---|
| Full cell | 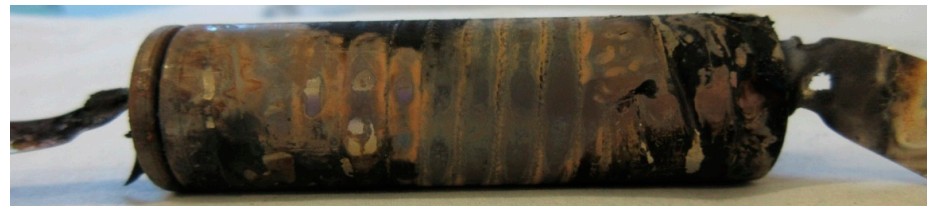 |
| Positive side | 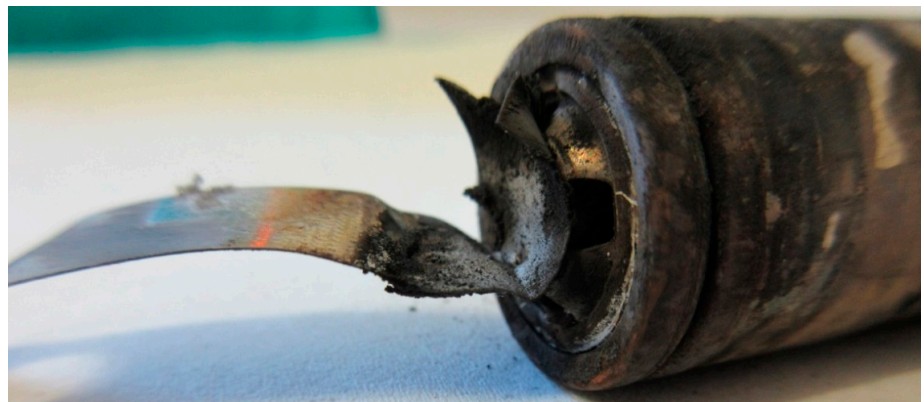 |
| Negative side | 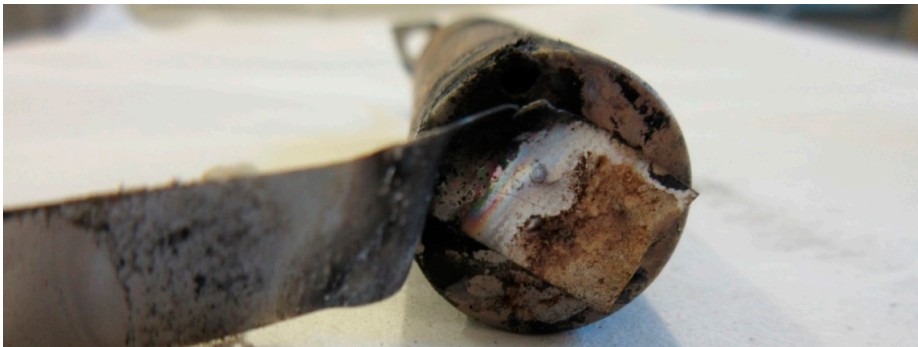 |

**Table A2.** Thermally abused LG HG2 at 50% SOC.

| | |
|---|---|
| Full cell | 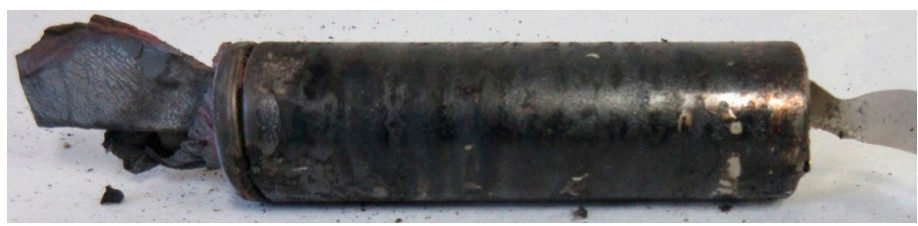 |
| Positive side | 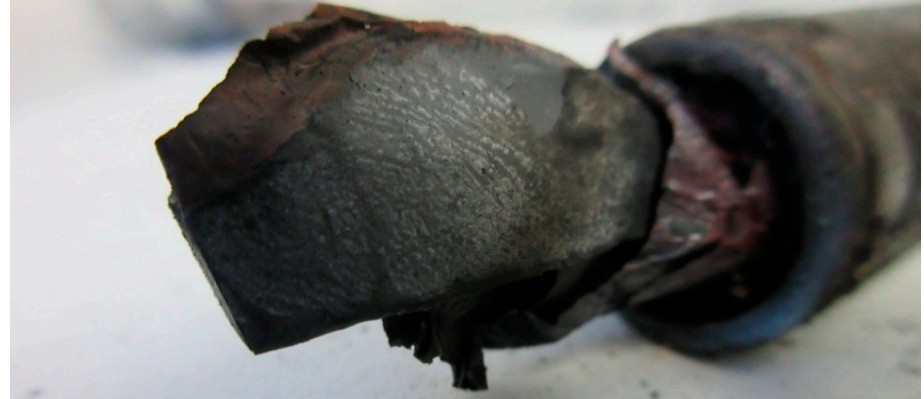 |
| Negative side | 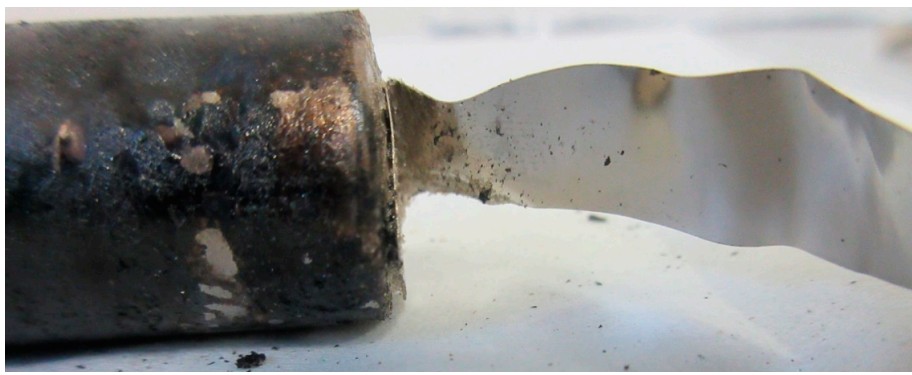 |

**Table A3.** Thermally abused Panasonic NCR GA at 100% SOC.

| | |
|---|---|
| Full cell | 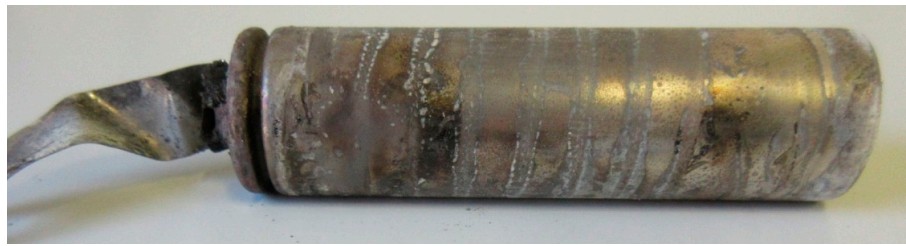 |
| Positive side | 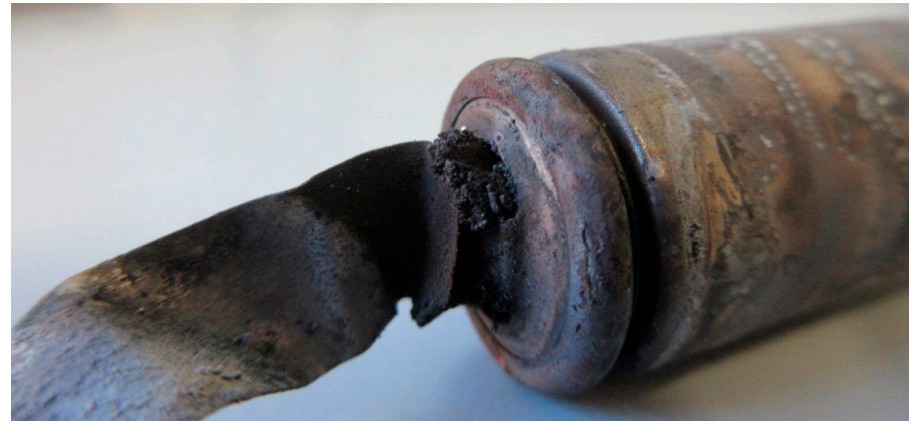 |
| Negative side | 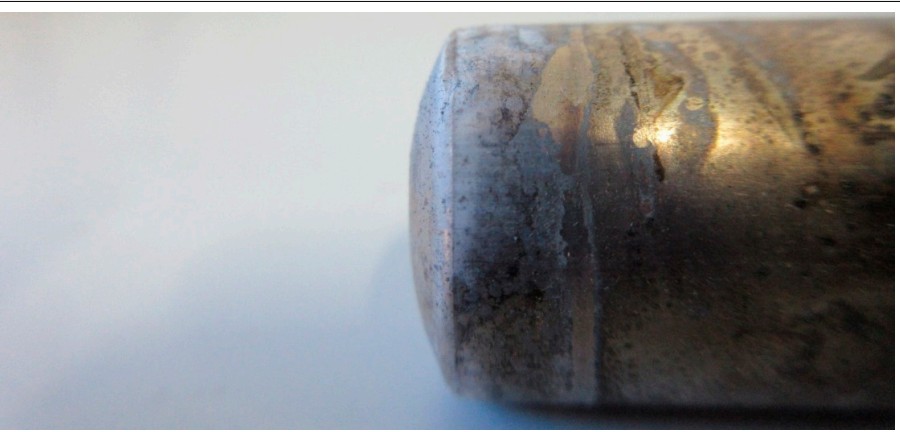 |

**Table A4.** Thermally abused Panasonic NCR GA at 50% SOC.

| | |
|---|---|
| Full cell | 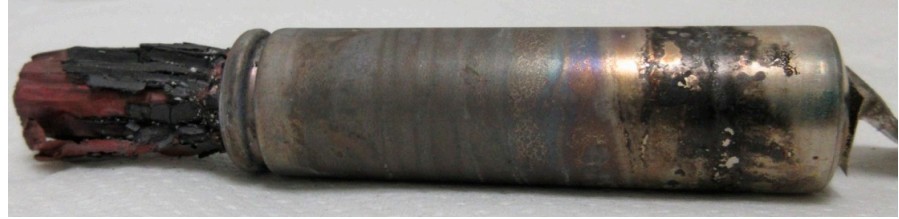 |
| Positive side | 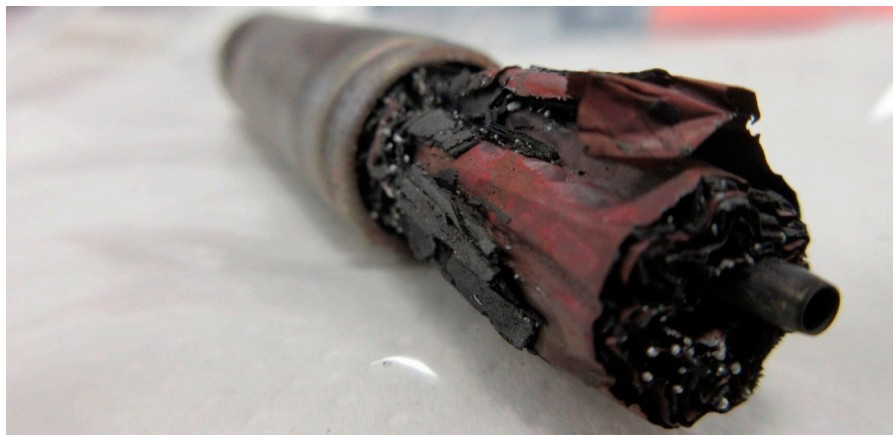 |
| Negative side | 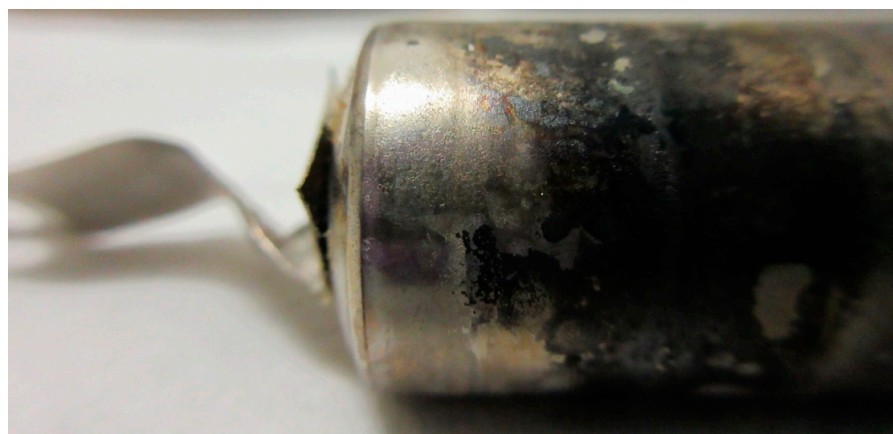 |

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
