# Peer review of "Understanding the Thermal Runaway of Ni-Rich Lithium-Ion Batteries"

_wevj, doi:10.3390/wevj10040079_

Round 1

Reviewer 1 Report

The manuscript approaches an interesting topic based on investigating thermal runaway behaviors of two cylindrical Ni-rich Li-ion batteries with different positive electrodes (NMC811 and NCA). Concerning to EV safety applications, this is a very relevant topic. However, same parts of the manuscript need to be rewritten/revised by the authors, namely the introduction and conclusion parts. It is my opinion that the manuscript has some lacks on scientific information and, in this way, needs to be minor revised prior to being accepted for publication.

In the following topics I stand out the main lacks that should be improved/revised on the manuscript:

In the abstract, quantitative results should be presented; In the introduction, the 4th and 5th paragraphs (line 66 up to 78) should be altered, because the same sentence is written in the abstract; Regarding the references: an article submitted to a journal should be consistent with the contents that it proposed in recent times. However, no references to “World Electric Vehicle Journal” papers are present. Authors should justify this point, or (it’s my suggestion) better study the bibliography of this journal and update the manuscript with appropriate works recently published. On line 135, please change “heat-wait-search” by “HWS”, it was already mentioned before in line 130. If possible, the labels of figure 3 should be increased. In the Technology selection, please specify which type of thermocouples were used (dimension, accuracy), and how were they fixed to the batteries surface. Please, clarify how many cells were tested in total, and if the results obtained were coherent for all cells. In the results and discussion, line 206, the statement “PP (polyethylene)” should be corrected. From the discussion provided about the temperature rise time detected in the figure 7, is not very clear for the readers the accuracy of these values for the two stages (1.2 °C/min in stage I and 48 °C/min in stage II). How was it calculated? The temperature axis in Figure 8 and 9 top, should be inserted to make readers easier to read. Line 327, please corrected “Figure 11-12” to “Figure 12-13”. The labels on Figure 14and 15 should be added. In the conclusion, should be addressed some sentences about quantitative results obtained during all the experimental tests. During all the manuscript, please correct the statement “Li-ion batteries” to “LIBs”.

Reviewer 2 Report

The lithium-ion battery is commonly used in many areas. And the safety control of the lithium-ion battery is of great importance. In this paper, the methodology of investigating the thermal runaway of pristine cells through experiment is proposed. The authors discuss the research experimental methodology, and the factors impacting the thermal runaway of selected technologies. The final results are positive for the future research. The following comments are made to the authors. (1) Normally, the abstract should be just one paragraph. (2) The related research work should be more comprehensive, especially in the introduction section. (3) In Line 183, P6, there is a redundant "[". (4) In Lines 244 and 245, P8, why the text is in italic type? (5) The conclusion section should be concise and highlighted the research results.

Reviewer 3 Report

Look at my all comments provided in the attachment and address them carefully.
